# DIFF4SPLAT: CONTROLLABLE 4D SCENE GENERATION WITH LATENT DYNAMIC RECONSTRUCTION MODELS

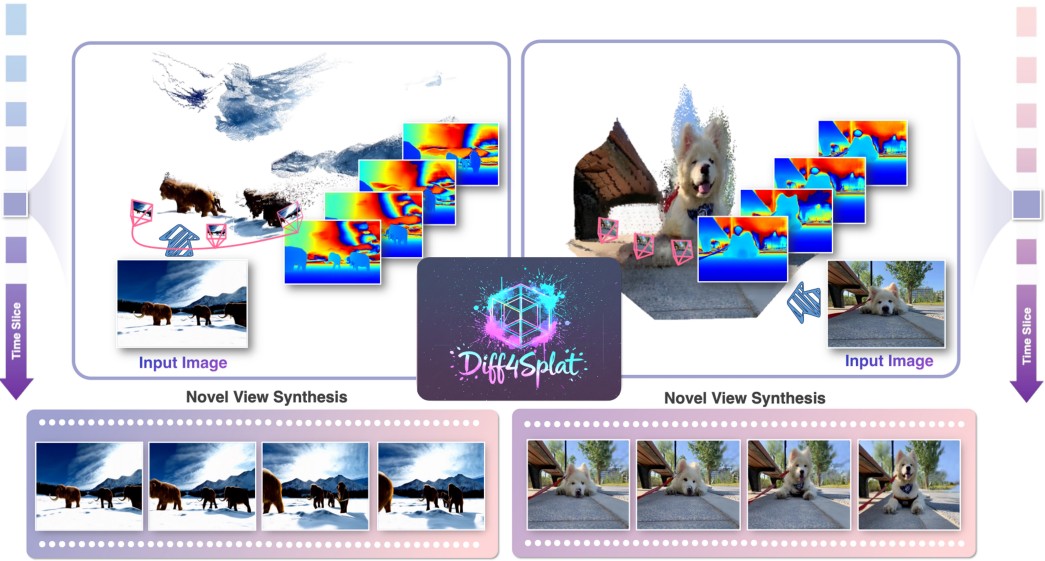

Figure 1: Given a single image, a specified camera trajectory, and an optional text prompt, our diffusion-based framework directly generates a **deformable 3D Gaussian field without test-time optimization**. The resulting representation supports diverse applications, including video generation, depth rendering, and novel view synthesis, enabling real-time rendering of dynamic scenes and interactive virtual exploration.

## ABSTRACT

We introduce **DIFF4SPLAT**, a **feed-forward method** that synthesizes controllable and explicit 4D scenes **from a single image**. Our approach unifies the generative priors of video diffusion models with geometry and motion constraints learned from large-scale 4D datasets. Given a single input image, a camera trajectory, and an optional text prompt, **DIFF4SPLAT** directly predicts a deformable 3D Gaussian field that encodes appearance, geometry, and motion, all in a single forward pass, without test-time optimization or post-hoc refinement. At the core of our framework lies a video latent transformer, which augments video diffusion models to jointly capture spatio-temporal dependencies and predict time-varying 3D Gaussian primitives. Training is guided by objectives on appearance fidelity, geometric accuracy, and motion consistency, enabling **DIFF4SPLAT** to synthesize high-quality 4D scenes in 30 seconds. We demonstrate the effectiveness of **DIFF4SPLAT** across video generation, novel view synthesis, and geometry extraction, where it matches or surpasses optimization-based methods for dynamic scene synthesis while being significantly more efficient. The code and pre-trained model will be released.

## 1 INTRODUCTION

Recent advances in monocular 4D reconstruction have shown promising results, yet their practicality is often limited by lengthy optimization (Wu et al., 2024b; Lei et al., 2024) or a lack of flexibility (Liang

et al., 2024c; Shen et al., 2025). Existing approaches to controllable 4D scene generation from a single image typically decompose the task into progressive video generation (Ren et al., 2025) followed by 3D neural reconstruction (Kerbl et al., 2023b; Yang et al., 2024b). While effective, these pipelines rely on multiple non-differentiable modules, require costly test-time optimization, or restrict themselves to static 3D scenes (Liang et al., 2024a) due to dataset limitations. More recently, the reliance on dynamic pointmaps in recent feed-forward methods (Zhu et al., 2025; Chen et al., 2025b) limits rendering quality, precluding photorealism and frequently introducing holes and artifacts. These challenges call for a unified and efficient framework that can directly generate dynamic 3DGS content.

We tackle the challenging task of single-stage controllable 4D scene generation from a single image, which requires simultaneous camera pose control, metric-scale geometry and motion prediction, and photorealistic rendering, all within a holistic deformable 3D particle-based representation (Yang et al., 2023). This problem is inherently ill-posed: it demands not only realistic image synthesis but also the recovery of dynamic geometry from sparse conditioning signals. The difficulty is further compounded by the scarcity of real-world video datasets with metric-scale depth. A successful solution must therefore combine photorealistic rendering with spatio-temporal coherence in the generated content. Such capabilities would unlock a wide range of applications, including immersive XR content creation, realistic environments for robotics, and scalable autonomous driving simulation.

As illustrated in Fig. 1, we aim to build a unified framework that directly predicts full 4D representations without test-time optimization or post-processing steps. This enables video generation, depth rendering, and novel view synthesis within a single diffusion model. To this end, we introduce **DIFF4SPLAT**, a holistic 4D diffusion transformer designed for scalable, data-driven scene generation. Addressing the scarcity of physically-grounded 4D data, we construct a large-scale annotation pipeline that converts real-world videos into spatio-temporal pointmaps with metric depth. To capture both visual and spatio-temporal dependencies, we extend the diffusion backbone with a **Latent Dynamic Reconstruction Model**, which transforms latent 2D tokens under temporal and camera embeddings. A lightweight prediction head then decodes these tokens into deformable 3D Gaussians, enabling real-time rendering of novel views and geometric maps such as depth. Our dataset provides rich supervision over appearance, geometry, and motion. Leveraging it, **DIFF4SPLAT** achieves state-of-the-art efficiency and geometric fidelity, while its unified representation yields significantly improved motion quality.

Our contributions can be summarized as follows:

- We propose **DIFF4SPLAT**, a unified diffusion-based model that directly generates deformable 3D Gaussians for controllable 4D scene synthesis.

- We construct a large-scale 4D dataset from synthetic and in-the-wild videos, annotated with appearance, metric-scale geometry, and motion.

- Extensive experiments demonstrate that **DIFF4SPLAT** produces high-fidelity 4D scenes from a single image, outperforming two-stage pipelines and existing camera-controlled video generation methods in both quality and efficiency.

## 2 RELATED WORK

**Video Diffusion Models** Video diffusion models (Ho et al., 2022) have demonstrated a remarkable capacity for generating high-quality, temporally coherent videos. Fine-grained control is typically achieved by adapting conditional image synthesis strategies (Zhang et al., 2023; Mou et al., 2024; Li et al., 2023b) to the video domain, incorporating diverse signals such as RGB images (Blattmann et al., 2023a; Xing et al., 2023; 2024a), depth maps (Xing et al., 2024b; Esser et al., 2023), motion trajectories (Yin et al., 2023; Niu et al., 2024), and semantic maps (Peruzzo et al., 2024). Despite these advancements, explicit camera motion control remains a relatively underexplored area. Existing approaches often rely on predefined motion categories (Guo et al., 2023; Blattmann et al., 2023a) or learnable LoRA modules (Hu et al., 2022). While methods like MotionCtrl (Wang et al., 2024b) employ camera extrinsics, they exhibit limited precision in complex scenarios, and MultiDiff (Müller et al., 2024) is constrained by class-specific training. More recently, several works (Xu et al., 2024; He et al., 2024; 2025) have leveraged Plücker coordinates (Sitzmann et al., 2021) for camera control, but still face challenges in producing realistic video outputs. Notably, the majority of current research

generates videos as 2D frame sequences, largely overlooking the joint generation of dynamic 3D representations (e.g., dynamic 3DGS).

**Static 3D Scene Generation**    Recent progress in generative models (Ho et al., 2020; Rombach et al., 2022b; Yang et al., 2025; Wang et al., 2025a) and 3D representations (Kerbl et al., 2023a; Mildenhall et al., 2020) has significantly advanced static 3D scene generation. One prominent research direction focuses on structured scene generation from layouts or graphs (Gao et al., 2024; Bai et al., 2023a; Po & Wetzstein, 2024; Vilesov et al., 2023; Yuan et al., 2025; Lin et al., 2025b; Lin & Mu, 2024; Lin et al., 2024a). Another line of research, more related to our work, addresses open-world scene generation from weak conditioning signals like text (Chung et al., 2023; Zhou et al., 2024) or images (Chung et al., 2023; Yu et al., 2024b; Liang et al., 2024a). These methods often rely on image diffusion models (Ho et al., 2020; Rombach et al., 2022b) as a backbone to provide strong 3D priors (Chung et al., 2023; Zhou et al., 2024; Yu et al., 2024b; Szymanowicz et al., 2025; Lin et al., 2025a; Wewer et al., 2024). The rise of video diffusion models has also motivated studies (Liang et al., 2024a; Liu et al., 2024; Yu et al., 2024c; Sun et al., 2024a) to leverage them for improved 3D-aware consistency. Our work distinguishes itself by pioneering dynamic scene generation, addressing the critical challenge of modeling motion.

**Dynamic 4D Scene Generation**    Static 3D generation methods are inherently limited to motionless scenes. The natural, albeit challenging, progression is dynamic 4D scene generation (Zhao et al., 2024b; Zhang et al., 2024; Chu et al., 2024; Liang et al., 2024d; Lin et al., 2025c; Li et al., 2024; Wang et al., 2025c; Zhu et al., 2025). Due to dataset limitations (Zhou et al., 2018a; Dai et al., 2017; Yeshwanth et al., 2023; Ling et al., 2024; Yu et al., 2023), prior works often tackle sub-problems. Some methods require a video and multi-view images of the first frame (Yu et al., 2024a; Wang et al., 2024a; Xie et al., 2024). Others generate 4D Gaussian Splatting from monocular video (Chu et al., 2025; Wu et al., 2024b; Liang et al., 2024d; Shen et al., 2025) or rely on costly per-scene optimization (Lei et al., 2024; Li et al., 2023c; Zhao et al., 2024a; Wang et al., 2025b; Wu et al., 2024a; Sun et al., 2024c). Recent feed-forward works generate dynamic pointmaps (Zhu et al., 2025; Chen et al., 2025b), but this kind of representation struggles to achieve photorealism, resulting in renderings with holes and artifacts. In contrast, our work introduces a generalizable method that generates an **explicit deformation Gaussian field** from a single image, without per-scene optimization.

## 3 METHODOLOGY

Our primary objective is the generation of a dynamic 4D scene representation from a single input image $\mathbf{I}_0 \in \mathbb{R}^{H \times W \times 3}$, text prompt $\mathbf{C}_{\text{ctx}}$, and the corresponding camera poses represented by Plücker embeddings (Jia, 2020) $\mathcal{P} \in \mathbb{R}^{T \times H \times W \times 6}$. As shown in Fig. 2, our methodology integrates a video diffusion model with a novel latent reconstruction Transformer. This unified framework synergistically combines 2D appearance priors, geometric constraints, and motion cues to synthesize high-fidelity 4D scenes. First, we leverage a pre-trained video diffusion model, conditioned on camera poses and the input image, to produce a video latent tensor $\mathbf{z} \in \mathbb{R}^{n \times h \times w \times c}$, where $n$ is the number of synthesized latent features, and $h, w, c$ denote the height, width, and channel dimensions of the latent features, respectively. We then introduce a Latent Dynamic Reconstruction Model (Sec. 3.2) that effectively integrates camera conditions with the generated latent features to predict a deformable Gaussian field, enabling rendering at novel viewpoints and time instances. Second, to facilitate dynamic scene generation, we augment the foundational static 3D Gaussian Splatting representation (Kerbl et al., 2023a) with an efficient mechanism for inter-frame deformation (Sec. 3.3). Third, we introduce a unified supervision scheme (Sec. 3.4) that incorporates photometric, geometric, and motion losses. Finally, we devise a progressive training strategy to ensure high-fidelity texture synthesis and enforce robust geometric constraints.

### 3.1 DATA CURATION

We start by developing a scalable 4D data annotation pipeline, meticulously designed to convert real-world videos into spatio-temporal point maps at metric scales. Our data curation strategy systematically integrates two complementary types of data sources:

❶ *Synthetic Datasets*: We leverage seven synthetic datasets: TartanAir (Wang et al., 2020), Matrix-City (Li et al., 2023a), PointOdyssey (Zheng et al., 2023), DynamicReplica (Karaev et al., 2023), Spring (Mehl et al., 2023), VKITTI2 (Cabon et al., 2020), and MultiCamVideo (Bai et al., 2025).

These datasets provide precise ground-truth annotations and controlled environmental variations, which are essential for learning robust geometric priors. ❷ *Real-world Datasets*: We incorporate two real-world datasets: RealEstate10K (Zhou et al., 2018b) and Stereo4D (Jin et al., 2025). These datasets offer authentic scene complexity and natural variations, which are crucial for enhancing the model's generalization capabilities. Inspired by (Zhu et al., 2025), we employ VideoDepthAny-thing (Chen et al., 2025a) and MegaSaM (Li et al., 2025b) to recover metric scale from these datasets, enabling more precise camera control within our generative framework (Bahmani et al., 2024).

Through this comprehensive data collection and processing pipeline, we amass approximately 130,000 high-quality 4D training scenes. Following a rigorous quality control protocol, which includes dynamic object masking and reprojection error filtering. We curate a refined dataset of approximately 100,000 synchronized multi-view videos, each annotated with metric point-maps and point motion trajectories. More technical details are available in Appendix B.

## 3.2 Latent Dynamic Reconstruction Model

While video diffusion models have demonstrated remarkable success in generating high-quality visual content, their direct application to synthesizing 3D-aware latents is non-trivial. This challenge arises from their inherent lack of explicit control over camera pose trajectories and their propensity to generate dynamic content that may lack the consistency required for robust 3D reconstruction. Drawing inspiration from recent advancements in latent-based diffusion models (Blattmann et al., 2023b; Rombach et al., 2022a; Pan et al., 2024; Liang et al., 2024b), we introduce the **L**atent **D**ynamic **R**econstruction **M**odel (**LDRM**), which significantly mitigates the computational overhead associated with per-scene optimization strategies. LDRM utilizes a pre-trained video diffusion model, conditioned on camera poses and an input image, to generate the latent tensor $\mathbf{z}$. The resulting video latents are inherently compact and 3D-aware, encapsulating a multi-view representation of the scene that is consistent in both structure and appearance, rendering them ideal for subsequent 3D lifting. Given the video latent tensor $\mathbf{z} \in \mathbb{R}^{n \times h \times w \times c}$ and the corresponding camera poses, we first transform these inputs into latent and pose tokens. Patchify modules ensure that both token sets possess identical sequence lengths. These token sets are then concatenated channel-wise and subsequently processed by a series of Transformer blocks (Ainslie et al., 2023). A lightweight decoding module regresses the attributes of 3D Gaussians from the Transformer's output tokens and uses a 3D deconvolutional layer to establish a pixel-level correspondence with the source video frames.

## 3.3 Deformable Gaussian Fields

A static 3D scene can be represented as a collection of $\mathbf{M}$ Gaussian primitives $\{\boldsymbol{G}_p\}_{p=1}^{\mathbf{M}}$. Each Gaussian $\boldsymbol{G}_p$ is characterized by its mean location $\boldsymbol{\mu}_p \in \mathbb{R}^3$, scaling factors $\boldsymbol{s}_p \in \mathbb{R}^3$, orientation quaternion $\boldsymbol{q}_p \in \mathbb{R}^4$, opacity $\alpha_p \in \mathbb{R}$, and color features $\boldsymbol{c}_p \in \mathbb{R}^C$. We use Spherical Harmonics (SH) to model view-dependent effects. The spatial influence of each Gaussian is given by:

$$\boldsymbol{G}_p(\mathbf{x}) := \exp\left(-\frac{1}{2}(\mathbf{x} - \boldsymbol{\mu}_p)^\top \boldsymbol{\Sigma}_p^{-1}(\mathbf{x} - \boldsymbol{\mu}_p)\right), \qquad (1)$$

where $\boldsymbol{\Sigma}_p$ is the covariance matrix derived from $\boldsymbol{s}_p$ and $\boldsymbol{q}_p$. Inspired by (Yang et al., 2024b; Lin et al., 2024b; Liang et al., 2025), we introduce a deformable 3D Gaussian formulation to represent dynamic scene. For each Gaussian $p$ at time step $t$, the predicted deformation field comprises a displacement for its mean, $\Delta\boldsymbol{\mu}_p^t \in \mathbb{R}^3$; an adjustment to its rotation, $\Delta\boldsymbol{q}_p^t \in \mathbb{R}^4$; and a modification to its scale, $\Delta\boldsymbol{s}_p^t \in \mathbb{R}^3$. The deformed parameters at time $t$ are updated as follows: $\boldsymbol{\mu}_p^t := \boldsymbol{\mu}_p^0 + \Delta\boldsymbol{\mu}_p^t$, $\boldsymbol{q}_p^t := \boldsymbol{q}_p^0 \otimes \Delta\boldsymbol{q}_p^t$ (quaternion multiplication), and $\boldsymbol{s}_p^t := \boldsymbol{s}_p^0 + \Delta\boldsymbol{s}_p^t$. These deformed Gaussians are then rendered using a differentiable Gaussian rasterization pipeline. Deformable Gaussian Fields is equipped with the LDRM, which generates a Gaussian feature map $\boldsymbol{G} \in \mathbb{R}^{(T \times H \times W) \times K_g}$, where $K_g$ denotes the number of parameters for each Gaussian primitive. Concurrently, the LDRM predicts a corresponding deformation map $\mathcal{D} \in \mathbb{R}^{(T \times H \times W) \times K_d}$. The dimensionality of this deformation, $K_d = 10$, comprises offsets for the mean ($\Delta\boldsymbol{\mu} \in \mathbb{R}^3$), rotation ($\Delta\boldsymbol{q} \in \mathbb{R}^4$), and scale ($\Delta\boldsymbol{s} \in \mathbb{R}^3$).

## 3.4 Training Objective

To enhance the geometric consistency of the generated latents, we introduce a progressive training scheme that jointly optimizes the network across multi-tasks via differentiable rendering.

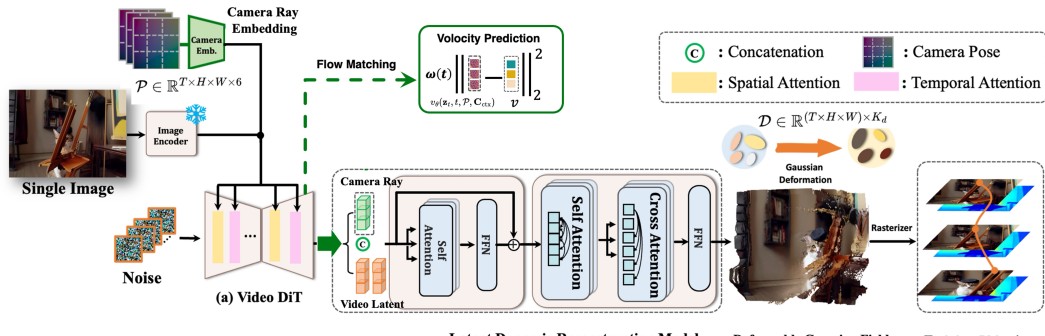

Figure 2: **Architecture of DIFF4SPLAT.** We present a high-fidelity dynamic 3DGS generation method from a single image through four key innovations: (1) video diffusion latents processed by our novel Transformer (Sec. 3.2), (2) a dynamic 3DGS deformation mechanism (Sec. 3.3), (3) unified supervision with photometric, geometric, and motion losses (Sec. 3.4), and (4) a progressive training scheme for robust geometry and texture.

**Flow Matching Loss** The Flow Matching (FM) (Lipman et al., 2023) approach learns the vector field that transports a noise distribution to the data distribution. Let $\mathbf{z}^{(0)}$ be a clean latent sequence from the data distribution $p_{\text{data}}$, and $\mathbf{z}^{(1)} \sim \mathcal{N}(0, \mathbf{I})$ be a sample from the prior gaussian noise. A probability path $\mathbf{p}_t(\mathbf{z}|\mathbf{z}^{(0)}, \mathbf{z}^{(1)})$ connects these samples, typically via linear interpolation $\mathbf{z}_t = (1-t)\mathbf{z}^{(0)} + t\mathbf{z}^{(1)}$ for $t \in [0, 1]$. The corresponding target vector field is $u_t(\mathbf{z}_t|\mathbf{z}^{(0)}, \mathbf{z}^{(1)}) = \mathbf{z}^{(1)} - \mathbf{z}^{(0)}$. Our model, $v_\theta(\mathbf{z}_t, t, \mathcal{P}, \mathbf{C}_{\text{ctx}})$, is trained to approximate this vector field by minimizing:

$$\mathcal{L}_{\text{FM}}(\theta) = \mathbb{E}_{t, \mathbf{z}^{(0)}, \mathbf{z}^{(1)}, \mathcal{P}, \mathbf{C}_{\text{ctx}}} \left[ w(t) \| v_\theta(\mathbf{z}_t, t, \mathcal{P}, \mathbf{C}_{\text{ctx}}) - (\mathbf{z}^{(1)} - \mathbf{z}^{(0)}) \|_2^2 \right], \tag{2}$$

where $w(t)$ is a weighting function for different noise levels and conditioning information (text prompt $\mathbf{C}_{\text{ctx}}$ and Plücker embeddings $\mathcal{P}$) is incorporated into $v_\theta$.

**Photometric Loss** To facilitate high-quality novel view synthesis, we optimize the 3DGS parameters using a composite loss:

$$\mathcal{L}_{\text{photo}} = \texttt{MSE}(\hat{\mathbf{I}}^k, \mathbf{I}^k) + \lambda_p \cdot \texttt{LPIPS}(\hat{\mathbf{I}}^k, \mathbf{I}^k), \tag{3}$$

where $\hat{\mathbf{I}}^k$ is the rendered image for view $k$, $\mathbf{I}^k$ is the ground-truth image, and $\lambda_p$ is a balancing coefficient for the LPIPS (Zhang et al., 2018) term.

**Geometric Loss** Inspired by (Li et al., 2025a), we introduce a geometric regularization term to enforce accurate depth relationships. Let $\hat{D}_k$ be the rendered depth map and $D_k^*$ be the ground-truth depth for view $k$.

$$\mathcal{L}_{\text{geo}}(\hat{D}_k, D_k^*) = 1 - \frac{\texttt{Cov}(\hat{D}_k, D_k^*)}{\sqrt{\texttt{Var}(\hat{D}_k)\texttt{Var}(D_k^*)}}, \tag{4}$$

where Cov and Var are covariance and variance functions. We also apply a total variation loss, $\mathcal{L}_{\text{TV}} = \|\nabla \hat{D}_k\|_1$, to enforce local smoothness.

**Motion Loss** Given 3D point tracking data, the ground-truth motion for a point $j$ is its displacement $\Delta \mathbf{x}_j$. The motion loss is:

$$\mathcal{L}_{\text{motion}} = \frac{1}{|\mathcal{O}|} \sum_{j \in \mathcal{O}} \left( \lambda_m \|\Delta \hat{\mathbf{x}}_j - \Delta \mathbf{x}_j\|_2 + \|\Delta \hat{\mathbf{x}}_j\|_1 \right), \tag{5}$$

where $\mathcal{O}$ is the set of tracked points, $\Delta \hat{\mathbf{x}}_j$ is the predicted displacement, and $\lambda_m$ is a weighting coefficient.

**Progressive Training Scheme**  To bridge the domain gap between video latents and the 3DGS representation, we introduce a three-stage progressive training scheme.

❶ **Static Geometry Pre-training (40K iterations).** We first establish a strong geometric prior by training LDRM on static scenes (e.g., TartanAir, RealEstate10K) at a low resolution ($256 \times 256$), using only photometric and geometric losses. During this stage, the deformation module (an 8-layer DPT head) is frozen.

❷ **High-Resolution Refinement (40K iterations).** With the deformation module still frozen, we enhance reconstruction fidelity by training on static scenes under a high resolution ($512 \times 512$).

❸ **Dynamic Scene Fine-tuning (20K iterations).** Finally, we unfreeze and fine-tune *the entire model* on dynamic datasets (PointOdyssey, DynamicReplica, Spring, VKITTI2, and Stereo4D). This stage employs the complete loss function, including a motion loss term, to learn temporal deformations. This progressive strategy, combined with our large-scale 4D dataset, enables our model to learn complex dynamics and generate high-fidelity, temporally coherent 4D scenes.

## 4 EXPERIMENTAL EVALUATION

### 4.1 IMPLEMENTATION DETAILS

Our framework builds upon a pretrained Video Diffusion Transformer model, CogVideoX (Yang et al., 2024a), operating within the latent space of a 32-channel $4 \times 8 \times 8$ compression 3D Causal Variational Autoencoder. The architecture comprises 32 blocks with a hidden dimensionality of 4096, specifically designed for image-to-deformation Gaussian field generation. Our LDRM architecture is composed of 16 standard Transformer blocks, the latent features have a channel dimension of $c = 32$ and are projected into a 64-dimensional embedding space before being processed by the Transformer backbone. To enable text control capabilities, each DiT block incorporates a cross-attention layer that integrates image embedding information from the T5 model (Raffel et al., 2020). For training, we employ the AdamW optimizer (Loshchilov & Hutter, 2019) with an initial learning rate of $10^{-5}$ and a weight decay of $10^{-4}$. The loss weighting hyperparameters are set to $\lambda_p = 0.5$ for the photometric loss and $\lambda_m = 2$ for the motion loss. A cosine learning rate scheduler is utilized, and the model is trained for 100,000 iterations until convergence. This training process requires approximately 7 days on a setup of 32 A100 GPUs, using BF16 mixed precision. At inference time, our Deformable Gaussian Diffusion model generates a complete 4D scene in **30 seconds**.

### 4.2 EVALUATION PROTOCOL

**Baselines**  We compare our holistic pipeline against the two-stage pipeline, which incorporates state-of-the-art techniques. Specifically, for this two-stage approach, we use AC3D (Bahmani et al., 2024) for single-image controllable video generation and Mosca (Lei et al., 2024) for dynamic Gaussian reconstruction. For comprehensive evaluation of camera controllability, we generated **160 evaluation samples** by applying five distinct camera trajectories (spiral, forward, backward, upward, and downward) to 32 unique text-captioned scenes.

**Metrics**  Our evaluation encompasses both prompt-scene consistency and aesthetic quality through: CLIP similarity score (Radford et al., 2021), Aesthetic score (CLIP-Aesthetic) (Schuhmann, 2023), VLM-based visual scorer Q-Align (QA-Quality) (Wu et al., 2023), and video quality metrics: FVD (Unterthiner et al., 2019) and KVD (Unterthiner et al., 2018). For geometric integrity assessment, we employ the MASt3R (Leroy et al., 2024) algorithm for local correspondence matching between input views and generated novel views and provide metrics through: Average matching correspondences, subject consistency score, and background consistency score (Zheng et al., 2025). More details are provided in Appendix D.

### 4.3 QUANTITATIVE AND QUALITATIVE EVALUATION

**Quantitative Results**  As shown in Tab. 1 and Tab. 2, our approach achieves competitive or superior performance across a variety of evaluation metrics. In terms of video generation and aesthetic quality (Tab. 1), our method delivers highly competitive results. Moreover, it significantly reduces reconstruction time to approximately 30 seconds. It offers a substantial efficiency improvement over methods like "AC3D + Mosca", which require around **45 minutes**, while maintaining strong

Table 1: Quantitative comparison on appearance fidelity and aesthetic quality. † indicates that this method requires per-scene optimization. Best results are in **bold**.

| Method | Video Generation & Aesthetic Quality | | | | | Rec. Time ↓ |
|---|---|---|---|---|---|---|
| | FVD ↓ | KVD ↓ | CLIP-Score ↑ | CLIP-Aesthetic ↑ | QA-Quality ↑ | |
| *Camera-Controlled Video Generation* | | | | | | |
| CameraCtrl (He et al., 2024) | 478.192 | 8.105 | 19.365 | 2.965 | 1.894 | 20s |
| AC3D (Bahmani et al., 2024) | 339.431 | 6.342 | 20.673 | 3.324 | 2.158 | 28s |
| *Explicit 3DGS Representation* | | | | | | |
| AC3D + Shape of Motion† (Wang et al., 2025b) | 373.045 | 6.511 | 16.201 | 3.043 | 1.838 | 18min |
| AC3D + SaV† (Sun et al., 2024b) | 327.122 | 5.816 | 19.018 | 4.371 | 2.382 | 35min |
| AC3D + Mosca† (Lei et al., 2024) | 235.961 | **2.012** | 20.214 | 4.999 | **2.842** | 45min |
| **Ours** | **210.153** | 2.316 | **23.123** | **5.231** | 2.813 | **30s** |

Table 2: Quantitative comparison on geometric integrity and reconstruction time. † indicates that this method requires per-scene optimization. Best results are in **bold**.

| Method | Geometric Integrity | | | Rec. Time ↓ |
|---|---|---|---|---|
| | Avg. Matches ↑ | Subject Consistency Score ↑ | Background Consistency Score ↑ | |
| *Camera-Controlled Video Generation* | | | | |
| CameraCtrl (He et al., 2024) | 2015.82 | 72.25 | 74.53 | 20s |
| AC3D (Bahmani et al., 2024) | 2489.16 | 75.64 | 75.91 | 28s |
| *Explicit 3DGS Representation* | | | | |
| AC3D + Shape of Motion† (Wang et al., 2025b) | 2874.22 | 83.13 | 83.33 | 18min |
| AC3D + SaV† (Sun et al., 2024b) | 3035.43 | 85.96 | 84.23 | 35min |
| AC3D + Mosca† (Lei et al., 2024) | 4500.68 | **86.23** | 90.43 | 45min |
| **Ours** | **5114.22** | 88.32 | 89.89 | **30s** |

geometric fidelity. As illustrated in Tab. 2, our method enables precise and camera-controllable generation with consistent geometric integrity.

**Qualitative Results**  As presented in Fig. 3, further highlight the advantages of Deformable Gaussian Diffusion. Our method generates 4D scenes that are visually more appealing, with greater temporal coherence and more accurate preservation of object structure and motion details than baseline methods. For example, our generated videos exhibit smoother transitions and fewer artifacts in dynamic regions compared to SaV and Mosca. This visual superiority stems from our model's direct prediction of deformable 3D Gaussians, which provides a rich and continuous representation of the scene's evolution over time, effectively capturing complex dynamics from a single image input. The dynamic motion generation capabilities of AC3D (Bahmani et al., 2024) and CameraCtrl (He et al., 2024) are inherited from their underlying 2D video DiT priors, often resulting in videos with limited dynamism.

**Generation Controllability**  Another key advantage of generating an **explicit** scene representation is the ability to ensure physical consistency through deterministic video "rendering from the input camera path". We validate this by quantifying camera pose fidelity. As shown in Table 3, we compare our method against AC3D using the Relative Pose Error (RPE) metric (Sturm et al., 2012) on our evaluation dataset, demonstrating a significant improvement in pose accuracy.

### 4.4 ABLATION AND ANALYSIS

**Effect of Deformation Gaussian Field**  Fig. 4 illustrates the importance of the deformation Gaussian module. Without this module, the model struggles to differentiate between camera movement and the motion of foreground objects. This inability to properly combine 3D Gaussian splats from different timestamps leads to motion blur, spike artifacts, and a general degradation in image quality. By employing the deformation Gaussian field, our model effectively fuses reconstruction information from various moments, thereby achieving higher visual quality.

**Effect of Explicit Representation**  Our adoption of an explicit 3D Gaussian Splatting representation offers several key advantages over implicit models, as detailed in Table 3. Firstly, it enables superior

Table 3: This comparison of the Average Relative Pose Error (RPE) highlights our method's superior performance over the implicit model, demonstrating enhanced accuracy in translation and rotation.

| Method | Avg. RPE (Translation) ↓ | Avg. RPE (Rotation) ↓ | Novel View Synthesis | Depth Rasterization | Real-time Interaction |
|---|---|---|---|---|---|
| Implicit 3D Models | 3.001 | 0.810 | ✓ | ✗ | ✗ |
| Ours (Explicit 3D Representation) | **0.012** | **0.008** | ✓ | ✓ | ✓ |

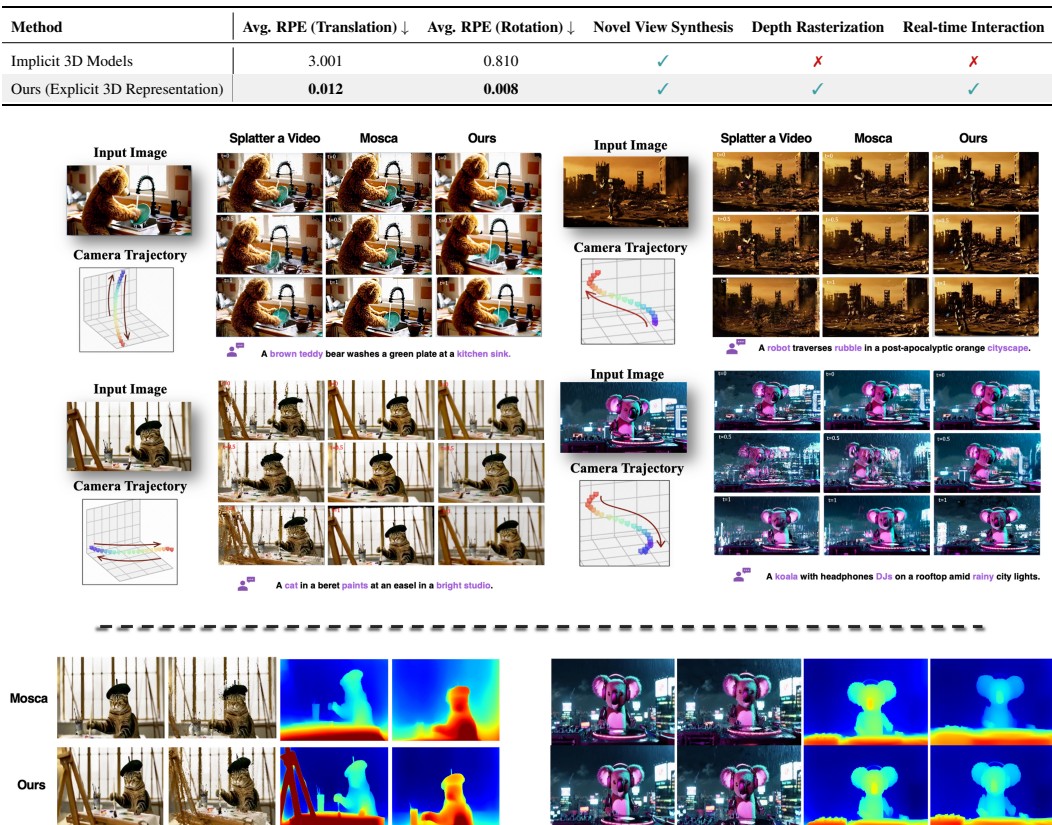

Figure 3: **Qualitative comparison with state-of-the-art methods.** DIFF4SPLAT (last column) generates more visually appealing and temporally consistent 4D scenes with superior geometric fidelity compared to baselines. Kindly zoom in for details.

camera controllability, drastically reducing the Relative Pose Error (RPE) in both translation and rotation. This ensures that the generated video faithfully adheres to the specified camera path. Secondly, the explicit nature of the representation unlocks additional functionalities not available in the implicit baseline, such as depth rasterization and real-time interaction. This not only enhances the model's utility but also provides greater flexibility for downstream applications.

**Effect of Motion Loss** While photometric, geometric, and flow matching losses are prevalent techniques in 3D generation (Liang et al., 2024a), we conduct a detailed ablation study on the components of our proposed motion loss. The quantitative results, presented in Table 4, demonstrate its efficacy. Specifically, ablating the motion loss component prevents the network from accurately modeling temporal deformations, which is critical for dynamic video synthesis. The absence of this loss significantly degrades the quality of scene reconstruction and negatively impacts all quantitative evaluation metrics.

**Effect of Progressive Training** Direct dynamic training without static pretraining. We observe that omitting the static pretraining phase and directly engaging in dynamic training leads to a failure in the initialization of static 3DGS. This, in turn, results in unstable training dynamics and ultimately compromises the quality of the generated 4D scenes. Progressive training, starting with a static scene understanding, provides a robust foundation, ensuring stable 3DGS initialization and facilitating the subsequent learning of complex dynamic elements, thereby significantly enhancing the overall performance and visual fidelity. Direct dynamic training will converge to a suboptimal state or require significantly more training time (e.g., 21 days versus 7 days) for progressive training to reach a similar baseline quality. As illustrated in Figure 5, after 100K training iterations, our progressive

Table 4: **Ablation Study on Motion Loss.** We evaluate the impact of our proposed motion loss on dynamic video generation.

| Method | FVD ↓ | KVD ↓ | QA-Quality ↑ | Avg. Matches ↑ | Subject Consistency Score ↑ | Background Consistency Score ↑ | Rec. Time ↓ |
|---|---|---|---|---|---|---|---|
| w/o motion loss | 351.382 | 3.351 | 2.145 | 4821.56 | 82.45 | 85.12 | 30s |
| Ours | **210.153** | **2.316** | **2.813** | **5114.22** | **88.32** | **89.89** | **30s** |

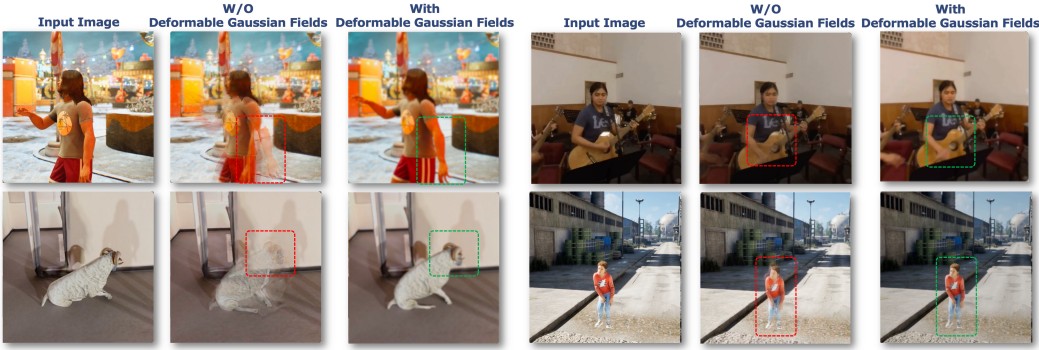

Figure 4: Ablation of the **Deformation Gaussian Field** shows that removing this module (the red bounding boxes) results in ghosting artifacts, particularly in the large motion frames.

training strategy yields significantly higher visual quality than direct dynamic training. This result underscores that progressive training not only enhances final performance and visual fidelity but also achieves superior results within the same computational budget, highlighting its resource efficiency.

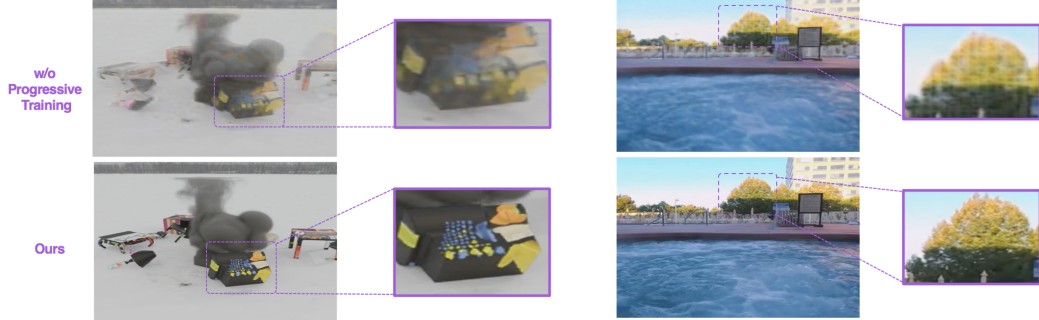

Figure 5: Ablation on the progressive training strategy.

## 5 CONCLUSION

In this work, we present a novel framework for explicit deformation Gaussian field generation from a single image in a *feed-forward* manner and achieves three key innovations: (1) unified diffusion transformer architecture integrating dynamic scene modeling, (2) geometry-aware latent representation enabling efficient view synthesis, (3) real-time rendering pipeline supporting practical applications. Extensive experiments demonstrate that our method achieves state-of-the-art performance in both geometric fidelity and computational efficiency, while eliminating the need for costly test-time optimization. We believe this work opens new opportunities for controllable 4D content creation at scale, bridging the gap between generative models and physically grounded scene understanding.

**Limitations and Future Work**    While our method achieves superior performance and efficiency, video generation remains the computational bottleneck. This could be addressed through parallel inference or optimized denoising strategies. Future work will focus on extending temporal coherence modeling and material property prediction.

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

## A  DECLARATION OF LLM USAGE

During the writing of the manuscript, we utilized a Large Language Model (ChatGPT) as a writing assistant. The scope of its use was limited to **improving grammar, polishing sentences, and enhancing the clarity and fluency of this manuscript**. The method, claims, experimental results and conclusions are developed by the authors.

## B  DATASET CURATION.

As describe in **Section** 3.1, we construct a collection of 130,000 diverse videos featuring dynamic scenes captured by stationary cameras. Real-world datasets such as RealEstate10K only provide relative camera parameters estimated via COLMAP (Schonberger & Frahm, 2016), resulting in an unknown global scale. To address this, we re-estimate both metric depth maps and camera extrinsics using recent foundation models, Video Depth Anything (Chen et al., 2025a) and MegaSaM (Li et al., 2025b), to recover aligned geometry across frames.

---

**Algorithm 1** Metric Depth Reconstruction via Relative Depth Alignment

---

1: **Input:** RGB Image $I$, pre-trained DepthAnything model $\mathcal{F}_{DA}$, MegaSaM model $\mathcal{F}_{MS}$, metric depth oracle $\mathcal{P}_M$

2: **Output:** Dense and metrically-scaled depth map $D^*$

3: $D_{rel} \leftarrow \mathcal{F}_{DA}(I)$           ▷ Generate relative depth map
4: $\mathcal{S} \leftarrow \mathcal{F}_{MS}(I)$           ▷ Generate segmentation mask set
5: $\mathcal{A} \leftarrow \emptyset$           ▷ Initialize anchor point set

6: **for** each mask $M_i \in \mathcal{S}$ **do**
7:     $d_{gt,i} \leftarrow \mathcal{P}_M(M_i)$           ▷ Query ground-truth metric depth for the mask
8:     **if** $d_{gt,i}$ is a valid measurement **then**
9:        $V_i \leftarrow \{D_{rel}(u,v) \mid M_i(u,v) = 1\}$     ▷ Extract corresponding relative depth values
10:        $d_{rel,i} \leftarrow \mathrm{median}(V_i)$           ▷ Compute a robust representative value
11:        $\mathcal{A} \leftarrow \mathcal{A} \cup \{(d_{rel,i}, d_{gt,i})\}$           ▷ Add the pair to the anchor set
12:     **end if**
13: **end for**

14:           ▷ Estimate optimal scale and shift by solving the least-squares problem
15: $(s^*, t^*) \leftarrow \arg\min\limits_{s,t} \sum_{(d_{rel,i}, d_{gt,i}) \in \mathcal{A}} (s \cdot d_{rel,i} + t - d_{gt,i})^2$

16:           ▷ Apply the transformation to the full relative depth map
17: $D^* \leftarrow s^* \cdot D_{rel} + t^*$

18: **return** $D^*$

---

Table 5: **Training Datasets Statistics.** Overview of the datasets used for training **DIFF4SPLAT** at scale, highlighting their dynamic nature, multi-camera setups, depth annotations, tracking capabilities, and real-world applicability.

| Dataset | Dynamic? | Multi-camera? | Depth? | Tracking? | Real? | #Scenes | #Frames |
|---|:---:|:---:|:---:|:---:|:---:|---:|---:|
| TartanAir (Wang et al., 2020) | ✗ | ✗ | ✓ | ✗ | ✗ | 0.4K | 0.49M |
| MatrixCity (Li et al., 2023a) | ✗ | ✗ | ✓ | ✗ | ✗ | 4.5K | 0.31M |
| RealEstate10K (Zhou et al., 2018b) | ✗ | ✗ | ✗ | ✗ | ✓ | 70K | 6.36M |
| PointOdyssey (Zheng et al., 2023) | ✓ | ✗ | ✓ | ✓ | ✗ | 0.1K | 0.18M |
| DynamicReplica (Karaev et al., 2023) | ✓ | ✗ | ✓ | ✓ | ✗ | 0.5K | 0.26M |
| Spring (Mehl et al., 2023) | ✓ | ✗ | ✓ | ✗ | ✗ | 0.03K | 0.003M |
| VKITTI2 (Cabon et al., 2020) | ✓ | ✗ | ✓ | ✗ | ✗ | 0.1K | 0.03M |
| MultiCamVideo (Bai et al., 2025) | ✓ | ✓ | ✗ | ✗ | ✗ | 14K | 11M |
| Stereo4D (Jin et al., 2025) | ✓ | ✗ | ✓ | ✓ | ✓ | 74K | 14.8M |

# C  More Implementation Settings

**Reproducibility**  To facilitate reproducibility, we present our detailed experimental settings and evaluation metrics in Section 4.1. This section provides a comprehensive description of our implementation details. Moreover, **our source code and pre-trained models will be publicly available.**

## C.1  Video Transformer Denosing Details

**Details of Model Inputs**  The model is conditioned on a single source image and a predefined camera motion trajectory, such as spiral, forward, backward, upward, or downward. Accompanying this, a textual prompt is provided, which can either be automatically generated from the source image using a Multimodal Large Language Model (MLLM) (Bai et al., 2023b) or set to a generic high-fidelity description, for instance, "a scene with 4K ultra HD, surround motion, realistic tone, panoramic shot, wide-angle view, cinematic quality".

**Classifier-Free Guidance**  **C**lassifier-**F**ree **G**uidance (**CFG**) has emerged as a prevalent technique for balancing controllability and sample diversity in diffusion models. However, we observe that its uniform scaling mechanism inadvertently introduces "over-sharpening artifacts" in the final frames of generated orbital sequences. To mitigate this limitation, we introduce a cosine-based dynamic guidance schedule during the sampling of validation videos, formulated as:

$$\gamma(t) = 1 + \gamma_{\max} \cdot \left( \frac{1 - \cos\left(\pi \left(\frac{N-t}{N}\right)^5\right)}{2} \right) \tag{6}$$

where $\gamma_{\max}$ denotes the maximum guidance scale, $N$ represents the total number of inference steps, and $t$ is the current timestep. This adaptive scheduling progressively reduces guidance intensity in later denoising stages, effectively preserving temporal consistency while maintaining sample fidelity. In our experiments, we set the total number of inference steps $N = 30$ and the maximum guidance scale $\gamma_{\max} = 7.5$.

## C.2  Deformation Field Generation

To predict the per-Gaussian deformations, our **LDRM** employs a lightweight spatio-temporal network. The network takes as input a latent representation of the scene at a canonical time step, conditioned on a time embedding for the target frame $t$. The architecture extracts features at multiple spatial resolutions to effectively capture both local and global motion patterns. The final layer of the network is a convolutional layer with a kernel size of $1 \times 1$, which projects the high-dimensional features into the final deformation map $\mathcal{D}$. This map has a dimensionality of $K_d = 10$ channels, which directly correspond to the predicted mean displacement (3 channels), rotational delta quaternion (4 channels), and scaling adjustment (3 channels) for each Gaussian primitive. No activation function is applied to the output layers for displacement and scale, allowing for unbounded predictions. The output quaternion components are normalized to ensure they represent a valid rotation.

## C.3  Details of Progressive Training Scheme.

Our progressive training scheme's efficacy in decoupling static and dynamic scene components is empirically validated. Initially, the model trains exclusively on static scenes, learning to predict an **identity deformation**. In this stage, positional and scaling offsets $(\Delta\boldsymbol{\mu}_p^t, \Delta\boldsymbol{s}_p^t)$ converge to zero, and rotational deformations $(\Delta\boldsymbol{q}_p^t)$ approach the identity quaternion, yielding a static representation as canonical Gaussians remain untransformed. Dynamic scenes are introduced in a subsequent fine-tuning stage. This decoupling is enabled by our Gaussian deformation formulation:

$$\boldsymbol{\mu}_p^t := \boldsymbol{\mu}_p^0 + \Delta\boldsymbol{\mu}_p^t, \quad \boldsymbol{q}_p^t := \boldsymbol{q}_p^0 \otimes \Delta\boldsymbol{q}_p^t, \quad \boldsymbol{s}_p^t := \boldsymbol{s}_p^0 + \Delta\boldsymbol{s}_p^t. \tag{7}$$

This design inherently separates the prediction of the canonical scene structure $(\boldsymbol{\mu}_p^0, \boldsymbol{q}_p^0, \boldsymbol{s}_p^0)$ from its temporal evolution $(\Delta\boldsymbol{\mu}_p^t, \Delta\boldsymbol{q}_p^t, \Delta\boldsymbol{s}_p^t)$.

Table 6: **Capability Comparison.** An explicit 4D representation enables a wide range of functionalities not supported by standard 2D video generation models.

| Capability | AC3D (Implicit 3D Models) | Ours (Explicit 4D Repr.) |
|---|---|---|
| Novel View Synthesis | ✓ | ✓ |
| Depth Rasterization | ✗ | ✓ |
| Geometry Extraction | ✗ | ✓ |
| Real-time Interaction | ✗ | ✓ |
| Interactive exploration Latency ↓ | 28000 ms | **6.7 ms** (↓ 99.98%) |
| Avg. Matches ↑ | 2489.16 | **5114.22** (↑ 105.5%) |
| Subject Consistency Score ↑ | 75.64 | **88.32** (↑ 16.8%) |
| Background Consistency Score ↑ | 75.91 | **89.89** (↑ 18.4%) |
| Cycle-Consistency ↑ | 20.68 dB | **34.5 dB** (↑ 66.8%) |

## C.4 Details of loss function weighting

The loss weights ($\lambda_p = 0.5$, $\lambda_m = 2$) were determined empirically through a series of experiments on a validation set. We started with equal weights and adjusted them to ensure that the model did not prioritize one objective at the expense of others.

## D Evaluation Protocol

To comprehensively evaluate our model, we utilize a suite of established metrics, Specifically:

❶ **Fréchet Video Distance (FVD) and Kernel Video Distance (KVD)** (Unterthiner et al., 2018): These metrics evaluate the quality and temporal coherence of generated videos by measuring the distance between the feature distributions of real and generated video sets. Lower scores for both FVD and KVD indicate higher fidelity and better temporal consistency.

❷ **CLIP-Score** (Radford et al., 2021): This metric quantifies the semantic similarity between the generated video frames and the input text prompt. It leverages the joint text-image embedding space of the CLIP model, where higher scores signify better alignment between the generated content and the textual description.

❸ **CLIP-Aesthetic** (Schuhmann et al., 2022): We use a model built upon CLIP embeddings to predict the aesthetic quality of the generated content. This model is trained on datasets with human aesthetic ratings, and a higher score suggests a more visually pleasing result.

❹ **QA-Quality** (Wu et al., 2023): This refers to a Visual Question Answering (VQA)-based evaluation, where a LLaMA2-powered model is employed to assess the logical consistency and objective quality of the generated scenes. The model assigns a score on a range from 0 to 5, where a higher score indicates superior quality.

❺ **Temporal Consistency Metrics (Avg. Matches, Subject Cons. and Bg. Cons.)**: Inspired by Video-bench (Ning et al., 2023), to specifically measure temporal stability, we use metrics based on dense optical flow or feature matching. Avg. Matches quantifies overall frame-to-frame consistency. Subject Consistency Score and Background Consistency Score measure the stability of the foreground subject and the background, respectively, after performing segmentation. Higher values for these metrics indicate smoother and more coherent videos.

## E Implicit vs. Explicit 3D Representations

Our work targets 4D scene generation by producing an "explicit" 3D representation (e.g., dynamic 3DGS), which offers capabilities substantially exceeding those of 2D video models. As demonstrated in Table 6, and inspired by prior work such as CAT4D (Zhang et al., 2008), an explicit 3D representation is a critical advantage for applications that demand a concrete understanding of and interaction with the world, including robotics and AR/VR.

Explicit 3D representations serve as a **"memory module"**, ensuring the consistency of the generated scenes. Unlike video generation models that predict 2D frames sequentially, our approach inherently enforces 3D consistency by predicting a single, unified explicit representation. Furthermore, 4D consistency is ensured by a training objective calculated from rendering the deformed 3D Gaussian representation from multiple viewpoints and at various timestamps. As shown in Table 6, we generate videos depicting a full 360-degree camera rotation. The resulting scenes exhibit seamless looping, where the final frame aligns perfectly with the first, showing no discernible seams or drift. We quantitatively verify this strong temporal consistency by measuring the similarity between the first and last frames (a.k.a., Cycle-Consistency), achieving a PSNR of 34.5 dB.

## F   Feed-forward vs. per-scene optimization

Existing methods that produce explicit 3D outputs, rely on a time-consuming, post-hoc optimization process to reconstruct scenes from generated videos. For instance, DimensionX (Sun et al., 2024a) requires **1.3K GPU hours** to perform scene optimization from a single video. Even state-of-the-art 4D reconstruction algorithms like Mosca (Lei et al., 2024) require approximately **0.5 hours** to process one input video. The primary motivation of this work is therefore to unify these disparate stages into a single, efficient, feed-forward framework capable of generating a 4D representation in approximately **30 seconds**, achieving 60× acceleration. Our model is designed for efficiency and scalability, enabling dynamic scene reconstruction in a matter of seconds, which is a critical feature for many real-world applications where speed is essential.

Compared to per-scene optimization methods, our proposed approach achieves a substantial reduction in memory consumption during the reconstruction process, decreasing from 80GB to 25GB (a 3.2× reduction) in the same setting. This efficiency gain stems from the elimination of gradient computation requirements. Furthermore, we claim that the two approaches are not mutually exclusive. As explored in recent work like CAT4D (Wu et al., 2024b), efficient, end-to-end models can serve as an excellent initialization for optimization-based methods, significantly accelerating their convergence. This potential synergy further highlights that developing fast, feed-forward models is a valuable research direction.

In summary, considering both the reconstruction and rendering stages (e.g., maximum GPU memory), our approach remains competitive in terms of memory consumption compared to per-scene optimization methods.

## G   Failure Cases

As shown in Figure 6, our method can produce artifacts when rendering novel timestamps, especially from disparate viewpoints. This issue, common to related methods, stems from ambiguity in estimating temporal deformations when propagating 3D Gaussians from multiple reference frames.

**Motion Ambiguity.** Single-image-to-4D generation is an ill-posed problem, as one image can imply multiple plausible motions (e.g., a bird gliding vs. flapping). This ambiguity can lead to inaccuracies in the predicted deformation field and corresponding visual artifacts. Incorporating more explicit motion priors in future work could address this limitation.

**Out-of-Distribution Generalization.** Model performance may degrade on out-of-distribution inputs, such as novel object categories or abstract artistic styles, resulting in lower-quality geometry and motion. Exploring few-shot domain adaptation techniques presents a promising direction for enhancing model robustness.

## H   More Visual Results

We provide more visualization results of **Diff4Splat** in Figure 7, Figure 8, and Figure 9.

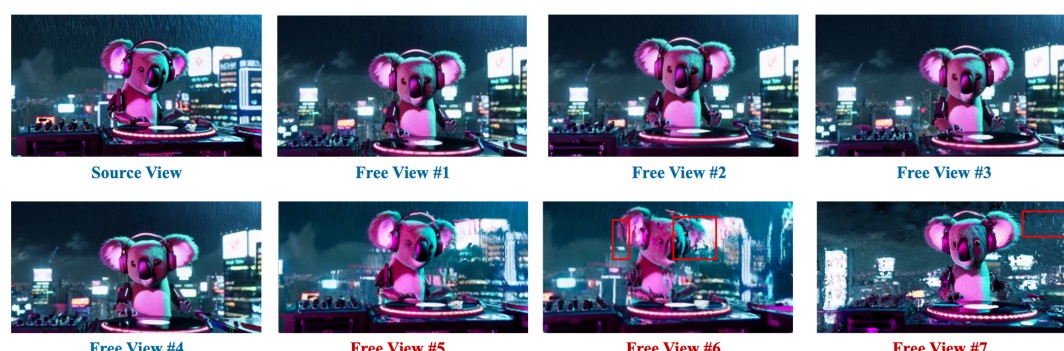

Figure 6: Failure Case. **DIFF4SPLAT** can produce artifacts when rendering novel timestamps, especially from disparate viewpoints. This issue, common to related methods, stems from ambiguity in estimating temporal deformations when propagating 3D Gaussians from multiple reference frames.

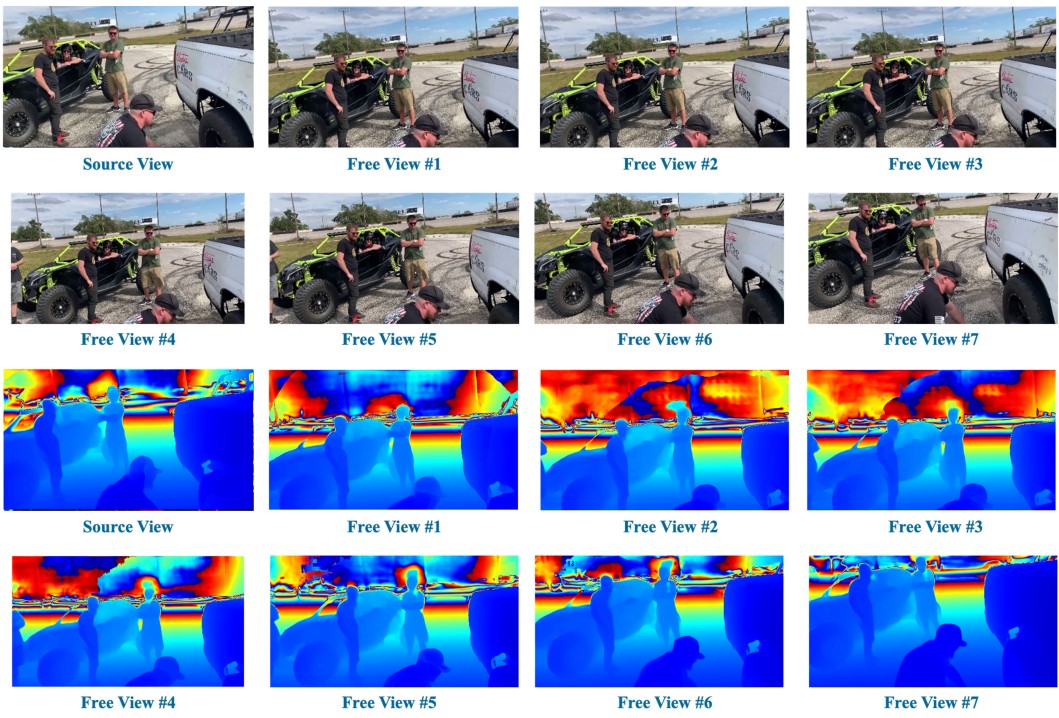

Figure 7: More qualitative of **DIFF4SPLAT** for 4D Scene generation.

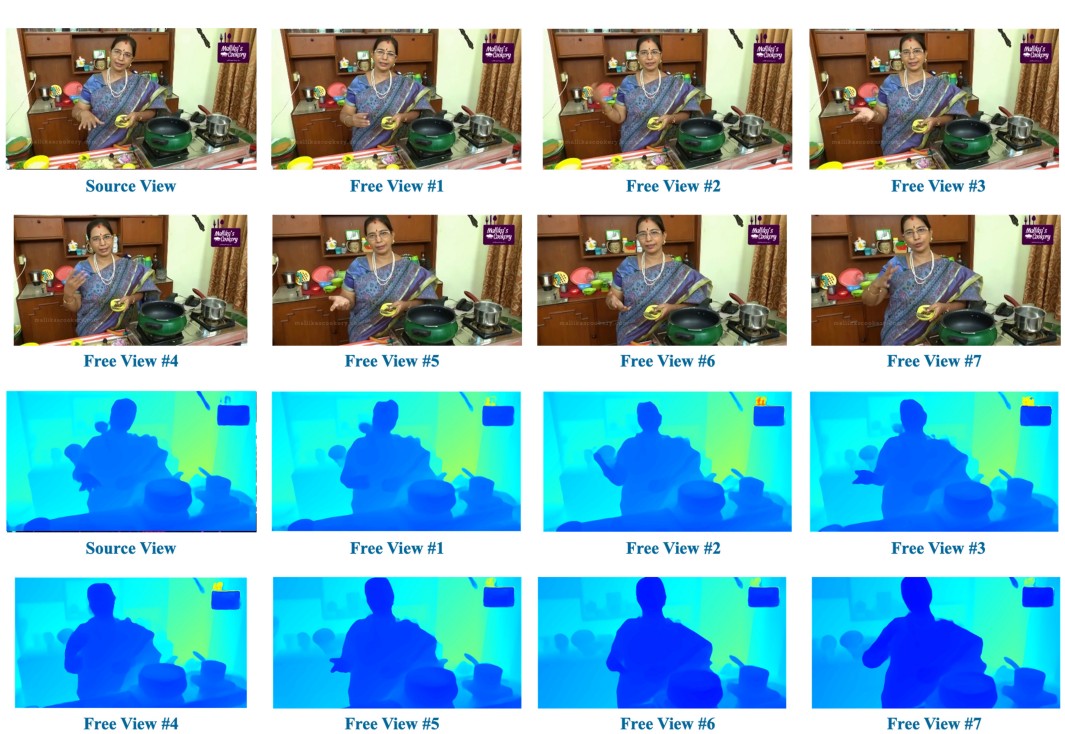

Figure 8: More qualitative of **DIFF4SPLAT** for 4D Scene generation.

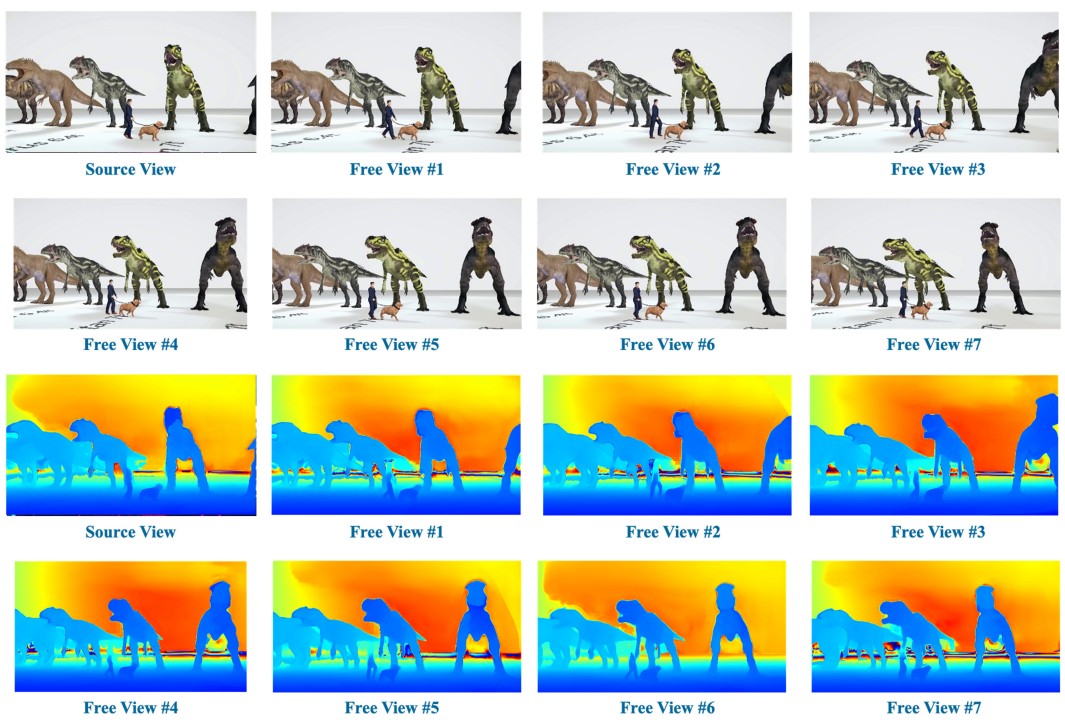

Figure 9: More qualitative of **DIFF4SPLAT** for 4D Scene generation.

