# OpenReview forum: "Diff4Splat: Controllable 4D Scene Generation with Latent Dynamic Reconstruction Models"
_ICLR.cc/2026/Conference — ICLR 2026 Conference Withdrawn Submission_

### Official Review · Reviewer_cHTH · 2025-10-19

**Soundness:** 3
**Presentation:** 2
**Contribution:** 3
**Rating:** 4
**Confidence:** 5

**Summary:**

The paper proposes Diff4Splat, a feed‑forward framework that predicts a deformable 3D Gaussian field directly from a single image, a user‑specified camera trajectory, and an optional text prompt, enabling controllable 4D scene generation without test‑time optimization. Video latant transformer augments video diffusion priors to jointly capture spatio temporal dependencies and produce time‑varying 3D Gaussian primitives, trained with objectives for appearance fidelity, geometric accuracy, and motion consistency. The system synthesizes high‑quality 4D scenes in roughly 30 seconds and demonstrates effectiveness across video generation, free‑viewpoint rendering, and geometry extraction.

**Strengths:**

1. Authors proposes a multi-stage training pipeline to support feed-forward 4D scene generation from a single image and claims to be the state-of-the-art in this domain.
2. Constructing a large 4D dataset annotated by appearance and geometric labels which would benefit the research community a lot to advance this field of 4D reconstruction / generation.
3. New Proposed architecture to predict deformable gaussians from video latents and this seems to drive the feed-forward nature of the method.

**Weaknesses:**

1. I feel the paper lacks adequate comparisons. One logical comparison would be use a single Image and feed it to a good image-to-video generation model like SORA-2 or other open source models and then use feed-forward 4D reconstruction methods like 4D Gaussian Transformer or other recent methods, and this way this method is also a fair comparison since the entire pipeline is feed-forward.
2. The results don’t seem to be very convincing. The final results seems to contain a lot of artifacts like the dog scene in the supplementary webpage. It seems to look like the depth prediction went bad which lead to floating artifacts.
3. Very few results: I feel tha since this paper is all about 4D scene generation, I would expect atleast 20-30 videos with different camera movements, i.e. same scene but different camera movements to see how well we could control the camera but I don’t see any examples in this direction. Even the examples shown are very bad and doesn’t exactly seem to represent the exact camera movement described like in Fig 3 (first & second row). I expect atleast the paper results to be of high quality.
4. The method also seems to generate deformations only for the specified view points, i.e. given certain camera views, it generates deformation for those particular view but this limits the very purpose of 4D controlloble generation. I would want to have an explicit 4D representation with a defomation network like 4DGS where once I have the weights, I can use that to render any view at any timestamp but your way of representation doesn’t allow that kind of rendering. If I change the camera poses in Diff4Splat, I will get a completely different scene altogether. I would assume the goal would be that, we can generate a 4D scene representation like 4DGS or Deformable 3DGS, where once we have the gaussian parameters and weights of the deformation network, we can render at any view point and any timestamp.
5. I also prefer if the steps taken in this research direction of 4D reconstruction are smaller. I feel that 4D scene reconstruction from images is itself a open problem and in that generating 4D scenes from few ques like single image or text prompt is harder problem to solve. Though I appreciate the authors bold move to tackle the issue of single image 4d generation, the current results tend to suggest that we are yet to solve other problems before solving this one. This is just a personal opinion and I feel extending the problem to a few images to 4D would be a more less ill-posed problem than single image 4d generation.
6. Also I feel the camera movement and object movement is very limited. In the video results - in 2nd and 3rd row, I see almost very minimal object motion and I think there is almost no motion in 2nd row. Examples like 1st row which have significant object motion tend to result in a lot of artifacts. Can the authors explain this as to why there is little object motion in the results and it would be great if the authors could show more different camera movements for the same scene
7. The literature review for this paper seems to be limited. I see papers like 4D Gaussian Transformer, PaintScene4D etc which do similar work to this paper but have failed to list them in the literature review.
8. Ablations isolating the video latent transformer design choices and each training loss (appearance/geometry/motion) are not described, hindering causal attribution.
9. How sensitive is the system to camera trajectory errors or calibration noise, and is there any self‑correction or bundling refinement built in ?

**Questions:**

I have highlighted the major questions in the weaknesses already, and I am summing them up below(I have summarised them shortly so that it is easier for the reviewer to quote the exact question they are answering. Please refer to the Weakness for the detailed problem and question asked):

1. Why have you not compared your method against a baseline where a single image is fed into a state-of-the-art image-to-video model (e.g., SORA-2 or open-source alternatives), followed by feed-forward 4D reconstruction methods such as 4D Gaussian Transformer? Wouldn’t such a pipeline serve as a fairer and more direct comparison since it is also fully feed-forward?

2. What could be causing the visual artifacts seen in several results (e.g., the dog scene in the supplementary webpage)? Do these stem from errors in depth prediction or another stage in your pipeline?

3. Why are there so few qualitative results? Given that this paper focuses on 4D scene generation, why not include at least 20–30 video examples demonstrating different camera movements for the same scene to test controllability?

4. In Fig. 3 (first and second rows), why do the rendered results not accurately match the described camera movements? Could you clarify how faithfully your method reproduces intended camera trajectories?

5. Since your approach seems to generate deformations only for specified viewpoints, how do you address the limitation that the same scene cannot be rendered consistently from novel viewpoints or timestamps?

6. Is there any plan to incorporate an explicit 4D representation (e.g., with a deformation network as in 4DGS) that allows rendering at arbitrary viewpoints and timestamps once the scene is learned?

7. Why is the object and camera motion so limited in your generated videos (e.g., almost no motion in the second row, minimal in the third)? Is this a limitation of your motion modeling or of the dataset used?

8. In examples with significant object motion (e.g., first row), why do noticeable artifacts appear? Could you analyze the trade-off between motion magnitude and reconstruction quality?

9. Given the current performance, do you believe single-image 4D scene generation is too ill-posed at this stage? Would extending the problem to “few-image to 4D” reconstruction be a more stable intermediate step?

10. Why were recent related works such as 4D Gaussian Transformer and PaintScene4D omitted from your literature review despite their conceptual similarity?

11. Can you provide ablation studies isolating the effects of your video latent transformer’s architectural choices and the individual training losses (appearance, geometry, and motion)?

12. How sensitive is your system to camera trajectory errors or calibration noise? Do you include any mechanism for self-correction or bundle adjustment to handle such inaccuracies?

---

### Official Review · Reviewer_xVDy · 2025-10-27

**Soundness:** 3
**Presentation:** 2
**Contribution:** 2
**Rating:** 4
**Confidence:** 4

**Summary:**

DIFF4SPLAT introduces a feed-forward, diffusion-based framework that generates controllable 4D dynamic scenes directly from a single image and prompts, based on camera trajectories. This method predicts a deformable 3D Gaussian field that represents both geometry and motion without the need for optimization during testing. The proposed Latent Dynamic Reconstruction Model integrates camera-conditioned video diffusion latents with temporal transformers to generate dynamic, explicit 3D Gaussian primitives. Trained on a large dataset combining synthetic and real-world sources, the system achieves high-quality scene synthesis in approximately 30 seconds per sample. It outperforms multi-stage pipelines, such as AC3D and Mosca, in terms of both visual quality and efficiency.

**Strengths:**

1. The paper presents a well-motivated, unified framework for single-image 4D scene generation, bridging video diffusion and dynamic 3D Gaussian representations. Its integration of dynamic deformation fields and camera-aware conditioning enables precise pose control and photorealistic rendering.
2. The progressive training scheme and multi-task loss design (photometric, geometric, motion) are thoughtfully engineered for stability and realism.
3. Quantitatively, DIFF4SPLAT matches or exceeds optimization-heavy baselines while being orders of magnitude faster, and qualitatively, it yields smoother motion and higher geometric fidelity.

**Weaknesses:**

. Many of the proposed components—such as the video-latent reconstruction model, progressive training—are adapted from prior works. The technical contribution thus feels incremental, focusing more on system integration than on conceptual innovation.

2. The reliance on the CogVideoX backbone makes the framework computationally expensive and limits its scalability for real-time or large-scale applications. The paper does not provide profiling results or efficiency analyses to substantiate its “ultrafast” claims relative to model complexity.

3. The related work and comparisons are not sufficiently comprehensive. Recent strong baselines such as 4Real-Video-V2 and other contemporary 4D diffusion approaches are not well discussed, leaving it unclear how DIFF4SPLAT differs from the current state of the art.

4. Qualitatively, the generated scenes exhibit limited motion range and modest viewpoint variation, making it difficult to assess the model’s true 3D consistency. Including depth visualizations would strengthen the geometric evaluation and provide clearer evidence of spatial coherence.

5. The training pipeline depends heavily on synthetic data and pseudo-metric annotations, which may restrict realism and generalization to unconstrained real-world videos. The paper would benefit from an analysis of how performance degrades when trained or tested on purely real data.

**Questions:**

A couple of questions;

1. Several components of DIFF4SPLAT resemble prior works in video-latent reconstruction and progressive training. Could the authors clarify what specific design choices or mechanisms make this framework technically distinct or more effective than previous systems?

2. The method relies on the CogVideoX backbone, which is computationally demanding. Have the authors profiled inference latency and memory usage?

3. How does DIFF4SPLAT differ in formulation or capability from such contemporaneous 4D diffusion systems, e.g. 4Real-Video-V2?

4. How sensitive is DIFF4SPLAT to errors or inaccuracies in camera poses? Does it degrade gracefully under moderate pose perturbations?

---

### Official Review · Reviewer_7G8y · 2025-10-30

**Soundness:** 3
**Presentation:** 3
**Contribution:** 2
**Rating:** 4
**Confidence:** 4

**Summary:**

The paper presents DIFF4SPLAT, a feed-forward framework that generates controllable 4D scenes from a single image. It combines video diffusion priors with geometry and motion constraints learned from large-scale 4D data. Given an image, camera path, and optional text prompt, it predicts a deformable 3D Gaussian field representing appearance, geometry, and motion — all in one forward pass without test-time optimization. A video latent transformer models spatio-temporal dependencies, enabling high-quality 4D scene synthesis in about 30 seconds.

**Strengths:**

1. The paper is clearly written.

2. It collects and organizes large-scale 4D datasets, which will be highly valuable for future research.

3. The experimental validation is thorough and comprehensive.

**Weaknesses:**

1. As shown in the results video, the range of camera motion is quite limited — most clips remain at a near-frontal viewpoint. The results are not impressive. In contrast, the training datasets (Real10K, DL3DV) contain much larger camera movements. What accounts for this discrepancy? Is it primarily constrained by GPU memory consumption?

2. Line 70 mentions that 4D content generation could support immersive XR content creation, realistic environments for robotics, and scalable autonomous driving simulation. However, given the weeknees1 above, it’s unclear what kinds of real applications can currently be expected from the presented results.

3. The overall technical approach is very similar to previous work — essentially an extension of Wonderland (CVPR 2025) to dynamic scenes. The main differences seem to be the use of a new dataset and the introduction of a motion-specific loss. While I understand that this research direction is extremely challenging and the achieved results are impressive, the overall contribution appears relatively limited.

**Questions:**

1. How many frames can the model generate? Does it need to match the frame length of the video diffusion backbone?

2. Do you consider the overall technical pipeline (4D + video diffusion latent + Gaussian prediction head) scalable in the long term? On one hand, such data are hard to obtain; on the other, the training cost is very high. Moreover, the current results still seem far from the ideal 4D field representation — being able to render any view at any time. Do you believe large-scale 4D content generation can realistically be achieved in the near future, and is this the right technical path toward that goal?

---

### Official Review · Reviewer_o5N6 · 2025-10-31

**Soundness:** 3
**Presentation:** 4
**Contribution:** 3
**Rating:** 4
**Confidence:** 4

**Summary:**

The paper proposes Diff4Splat, a single-stage, feed-forward model that turns one input image, a camera trajectory, and an optional text prompt into an explicit 4D representation: a deformable 3D Gaussian field that encodes appearance, geometry, and motion. The core is a Latent Dynamic Reconstruction Model (LDRM) that takes video-diffusion latents and camera embeddings to regresses time-varying 3D Gaussian primitives. The training follows a three-stage progressive pipeline (static pretrain → high-res refine → dynamic fine-tune). The method claims ~30s end-to-end 4D generation without test-time optimization and demonstrates improvements over camera-controlled video baselines and two-stage (video→4DGS) pipelines across quality and geometry metrics. The authors also curate a large mixed synthetic/real 4D dataset with metric depth/motion supervision via foundation models.

**Strengths:**

1.	Clear single-stage formulation with explicit 4D output. Predicting a deformable 3DGS directly (rather than first generating video) removes costly post optimization and enables novel view rendering and real-time interaction.

2.	The LDRM design (video-diffusion latents + camera embeddings + transformer + Gaussian/deformation heads) is precise. The deformation parameterization and the three-stage progressive training are explained with rationale and ablations.

3.	Tables show better FVD/KVD/CLIP-Aesthetic than baselines and much faster wall-clock vs. two-stage systems (e.g., AC3D+Mosca ~45 min vs. 30 s), alongside higher matching/consistency scores; interactive rendering latency (6.7 ms) is compelling.

4.	The metric-scale supervision pipeline (VideoDepthAnything + MegaSaM) and the ~100k curated 4D scenes if helpful for the community.

**Weaknesses:**

1.	The building blocks (diffusion transformer, Gaussian splatting, flow matching) are adaptations of existing methods; innovation lies more in integration than new theory.

2.	The paper acknowledges that generating 4D motion from a single image is an ill-posed problem, as one image can correspond to multiple plausible motions. While this is a fundamental challenge, the current framework seems to lack a mechanism for explicit control over the type of motion beyond what is implicitly learned. The text prompt appears to primarily control scene content rather than dynamics.

3.	While multiple metrics are used, the test scenarios focus mainly on synthetic and controlled camera paths; performance on in-the-wild or arbitrary trajectories remains unclear.

**Questions:**

1.	The LDRM directly regresses 3D Gaussian parameters from the video latent tensor. Could you provide more intuition on how the model effectively learns this complex 2D-to-4D mapping?

2.	The paper mentions an "optional text prompt" as input. To what extent can this text prompt control the dynamics of the scene? For instance, could you use a prompt to make an object move faster, slower, or in a different manner (e.g., "a bird flapping its wings" vs. "a bird gliding gracefully")?

3.	The evaluation shows impressive camera control on predefined trajectories (spiral, forward, etc.). How robust is the model to more complex or unconventional camera paths that may differ significantly from those in the training data?

4.	It is recommended to test the model's performance on 3D point tracking tasks, similar to the evaluation in Shape-of-Motion, to verify that the model learns proper motion.

---

### Note · Authors · 2025-11-12

I have read and agree with the venue's withdrawal policy on behalf of myself and my co-authors.